# Plasma Radiofrequency Ablation for Scar Treatment

**DOI:** 10.3390/jcm11010140

**Published:** 2021-12-27

**Authors:** Adone Baroni, Pasquale Verolino

**Affiliations:** 1Department of Dermatology and Venereology, University of Campania “Luigi Vanvitelli”, 80131 Napoli, Italy; 2Plastic Surgery Unit, University of Campania “Luigi Vanvitelli”, 80120 Napoli, Italy; pasqualeverolino@libero.it

**Keywords:** plasma energy, plasma radio frequency ablation, scar treatment

## Abstract

Scars are a common disfiguring sequela of various events such as acne, hidradenitis suppurativa, surgery, trauma, and burns, which can lead to serious psychosocial problems with a negative effect on the quality of life. Many conventional approaches have been proposed for the treatment of scars, including surgical techniques, dermabrasion, chemical peels, topical silicone gel, 5-fluorouracile and dermal fillers injection or autologous fat transfer for atrophic scars, and corticosteroids injection for hypertrophic and keloid scars; however, they have sporadic effects. Ablative lasers, such as carbon dioxide laser or Erbium Yag laser, are associated with many collateral effects limiting their application. Non-ablative laser treatments have been shown to be safer and to have fewer side effects, but they have a reduction of clinical efficacy compared to ablative lasers and a minimal improvement of scars. The demand for minimal invasive and safe technology for the treatment of a scars has stimulated the search for more effective novel therapy with fewer collateral effects. Plasma radiofrequency ablation is a new technique consisting of the generation of plasma energy through the production of ionized energy, which thermally heats tissue in a uniform and controlled manner, through a plasma radiofrequency device, inducing a sublimation of the tissue. The aim of this study is to evaluate the effectiveness of P-RF ablation in the treatment of scars performed with D.A.S. Medical device (Technolux, Italia), which is a tool working with the long-wave plasma radiofrequency principle.

## 1. Introduction

Scars are a common disfiguring sequela of various events such as acne, hidradenitis suppurativa, surgery, trauma, and burns, which can lead to serious psychosocial problems with a negative effect on the quality of life [1].

Skin tissue repair results in a broad spectrum of scar types, ranging from “normal” fine line scars to abnormal scars, including stretched, atrophic, hypertrophic, keloid scars, and scar contractures [1].

Abnormal healing processes can complicate the normal wound healing, resulting in the development of proliferative scars, such as hypertrophic scars and keloids, resulting from excessive deposition of collagen at sites of prior dermal injury or of wound repair due to an imbalance between collagen biosynthesis and matrix degradation [2]. Hypertrophic scars are raised scars that remain within the boundaries of the original lesions; however, keloid scars spread beyond the margins of the original lesions invading the surrounding normal skin [1].

Atrophic scars appear when dermal collagen and connective tissue production inadequately compensate for the tissue loss. They are depressed below the surrounding skin; small and round scars commonly arise after acne, hidradenitis suppurativa, or chickenpox.

Many conventional approaches have been proposed for the treatment of normal and abnormal scars, including surgical techniques, dermabrasion, chemical peels, topical silicone gel, 5-fluorouracil and dermal fillers injection or autologous fat transfer for atrophic scars, and corticosteroids injection for hypertrophic and keloid scars; however, they have sporadic effects [2,3]. For unresponsive scars or as alternative therapies, laser treatments may be effective [4]. Ablative lasers, such as carbon dioxide laser or Erbium Yag laser, are associated with delayed erythema, persistent hyperpigmentation, prolonged healing times, infections, and even worsening of scarring, which limits their application [4]. Conversely, non-ablative laser treatments have been shown to be safer and have fewer side effects, but they have a reduction of clinical efficacy compared to ablative lasers and a minimal improvement of scars. Ablative fractional lasers, such as CO_2_ fractional laser, have been shown to have some beneficial effects on atrophic scars with short healing times, short downtime, and duration of erythema, yet post-treatment collateral effects can also occur, especially for Fitzpatrick skin type III and IV [5].

Plasma radiofrequency ablation (P-RF) technique, recently used for the treatment of xanthelasma palpebrarum [6], full facial rejuvenation and photoaging [7], facial acne and fine lines [8], non-surgical blepharoplasty [9], and benign skin lesions removal [10], can be considered a good option to meet the need for less invasive and safe technology for the treatment of scars.

The demand for minimally invasive and safe technology for the treatment of scars has stimulated the search for more effective novel therapy with fewer collateral effects. Plasma radiofrequency (PRF) ablation is a new technique consisting of the generation of plasma energy through the production of ionized energy that thermally heats tissue in a uniform and controlled manner, through a plasma radiofrequency device, inducing a sublimation of the tissue [11]. In contrast to ablative treatments, such as lasers and traditional radiofrequency, plasma sublimation leaves a layer of intact and desiccated epidermis that acts as a natural biologic dressing, avoiding damaging the deeper layers of the skin and predisposing to better healing [10]. Moreover, the detached spots of the sublimation technique leave spared columns that further aid in healing, with a still more rapid recovery and consistent aesthetic results.

The aim of this study is to evaluate the effectiveness of P-RF ablation in the treatment of scars performed with D.A.S. Medical device (Technolux, Italia), a tool working with long-wave plasma radiofrequency principle.

## 2. Patients and methods

Following the receipt of the ethics waiver, 10 patients (six females and four males; 21–54 years; skin phototypes I–IV) affected by unsightly scars (five post-acne atrophic scars, three post-traumatic hypertrophic scars, two hidradenitis suppurativa atrophic/hypertrophic scars) were enrolled for longwave P-RF (LWP-RF) ablation treatments. Patients with scars younger than 2 years, or with keloids and burn scar contractures, were excluded from the study entry.

Prior to the treatments, all patients signed an informed consent form. No contraindications to treatment, such as Fitzpatrick skin types V–VI, presence of tanning, concomitant infection, diabetes, pregnancy, use of oral isotretinoin within the previous 6 months, presence of collagen vascular disorders or other autoimmune diseases, ablative skin procedures carried out within the previous 3 months, and malignancy, were present.

The scars to be treated (Figures 1A–3A) were anesthetized with topical anesthetic cream (EMLA, Astrazeneca s.p.a., Milano, Italy) without occlusion for 1 h before starting every treatment. The scars were disinfected with the antiseptic solution following the removal of the anesthetic cream with sterile gauze.

In patients affected by post-acne atrophic scars and hidradenitis suppurativa atrophic/hypertrophic scars, LWP-RF ablation was performed in four sessions (Figures 1B and 3B). Atrophic scars were treated at 0.6-watt energy and 3 Hz frequency with detached spots of sublimation performed on the edges around the scars, leaving columns of the intact epidermis and sparing the central atrophic part to break down the edges and to stimulate the rise of the central skin depression (Figures 1B and 3B). The hypertrophic parts of hidradenitis scars were treated with superimposed passes at 0.6-watt energy and 2 Hz frequency to break down the thick skin (Figure 3B).

Post-traumatic hypertrophic scars were fully ablated in three sessions with two superimposed passes at 0.8-watt energy and 2 Hz frequency with no columns of sparing tissue, to break down the thickening (Figure 2B). Once the scar was smoothened, the residual stretched appearance was ablated in further two sessions with only one pass at 0.6-watt energy and 3 Hz frequency with detached spots of sublimation to achieve the shrinkage (Figure 2B).

In all patients, the sessions of treatment were carried out at 40-day intervals, and the final endpoint for completion of the single session was moderate burnishing. After the LWP-RF procedure, they applied an antibiotic ointment for 10 days, avoiding sun exposure during the sessions and for 6 months after the last session.

## 3. Results

Treatment sessions and post-treatment period were well tolerated by all patients, which referred to only negligible concerns during the treatments. At the end of the sessions, only moderate burnishing was present; reepithelialization occurred within 7–10 days. All patients were followed up at 3 and 5 months from the last treatment, and the outcome was already visible at 3 months from the last treatment. No side effects, such as infections or dyschromia, occurred in any patient.

A significant aesthetical improvement of the scars was observed after four LWP-RF sessions of treatment for all patients, with great improvement of the skin texture, reduction of the visibility of the treated scars, and high patient satisfaction (Figure 1B, Figure 2B and Figure 3B). Visual analog scale (VAS) score was used to evaluate the patient satisfaction rate in a range from 0 to 10. All patients declared a VAS > 8.

## 4. Discussion

Over the past 35 years, the treatment of scars has undergone significant changes influenced by the development of new devices and technological evolution. The use of fractional lasers was an important step in scar therapy, and the advent of the high-energy pulsed CO_2_ laser in the 1990s provided an important new tool for the treatment of scars that depicted very high efficacy [12,13]. However, the prolonged downtime and high complication rates led to the adoption of fractional lasers [14]. Fractional resurfacing was highly innovative in the treatment of scars [15]. The ability to resurface directly into the dermis changed the ability to reshape the scars by either ablating or coagulating them, allowing the body’s repair mechanism to replace the lost, defective tissue [16]. Nonablative fractional resurfacing has limitations, such as high costs and adverse events that limit the procedures. Although fractionating the laser beam significantly reduced the incidence of hypo or hyperpigmentation (vs. nonfractional lasers), high energies led to aggressive treatments that produced hyperpigmentation. Nonablative and ablative fractional treatments are also associated with discomfort and downtime.

Picosecond lasers have been used successfully for scars, especially for treating darker skin types and avoiding post-inflammatory hyperpigmentation [17]. Another advantage of this modality is that there is very little downtime, allowing patients to continue their daily activities. Disadvantages of picosecond lasers are the need for multiple treatments and the high cost of the equipment.

The need to overcome the limits and difficulties posed by lasers has led to seeking alternative methods and tools that could make use of mechanisms other than laser technology.

Plasma is “the fourth state of the matter” derived from the ionization of neutral gases present in the air [6,7,8,9,10]. DAS medical device utilizes plasma energy created from the ionization of atmospheric gas between the device and the skin and consists of a hand piece with a sterile tip and a wrist-generator, allowing the adjustment of power and frequency values. It works without contact between the toe and the skin; the resulting plasma spark sublimates the superficial layers, immediately transferring the stored thermal energy to the skin surface, heating in a controlled uniform manner [6,7]. P-RF energy triggers microplasma sparks in the air between the tip of the device and the skin surface, producing mild epidermal ablation and perforating the dermis superficially with the spot of 1 mm diameter. The mild epidermal sublimation leaves a layer of intact and desiccated epidermis, avoiding damaging the deeper layers of the skin. In addition to a mechanical effect that shapes the surface on which it impacts, the detached spots of the sublimation technique induce a thermal effect that promotes skin regeneration and extensive dermal fibroblast remodeling, including new collagen synthesis and deposition, also stimulating a rapid reepithelialization.

One of the main features of action offered by this technique compared to fractionated lasers that act homogeneously on the entire treated area is to customize the treatment millimeter by millimeter by acting focally on reducing the thickening of the hypertrophic area of the scar and the edges of the peripheral area of the atrophic scar. Moreover, it also involves stimulating the depressed atrophic area of the scar, similar to fractional laser action, stimulating its ascent up to the level of the skin plane. Added to this is the lack of side effects related to the absence of a light source, the ability of the plasma effect to moderate the heat transfer, and consequently, the reduction in the inflammatory effect, inducing, however, an advantageous stimulation of the dermis.

The advantages of this technique include the lack of absolute contraindications, minimal intraoperative pain, quick treatment, rapid formation of the postoperative protection layer, fast healing of wound surface, immediate return to normal activity, and optimal results in the treatment of hypertrophic and atrophic scars.

Finally, the low cost of the equipment compared to laser devices results in an added value to the advantage of both the operator and the patient, who access an effective treatment at significantly reduced costs compared to laser treatment.

The limitations of the technique mainly concern the choice of the type of scar, as satisfactory results were not seen on keloids, so much so as to exclude patients with keloids from the study. The small number of patients recruited for P-RF-ablation treatments could represent a limitation of this study, therefore ensuring meaningful results; further clinical studies are needed on a larger number of patients to better determine the effects of this relatively new technique for the treatment of scars, alone or in combination with other kinds of treatments.

In conclusion, the trend towards combined therapy in the treatment of scars has stimulated planning for the near future to evaluate the results of the combination of P-RF ablation with non-ablative lasers in order to settle the result and further improve the general appearance of the treated area without incurring any side effects.

## Figures and Tables

**Figure 1 jcm-11-00140-f001:**
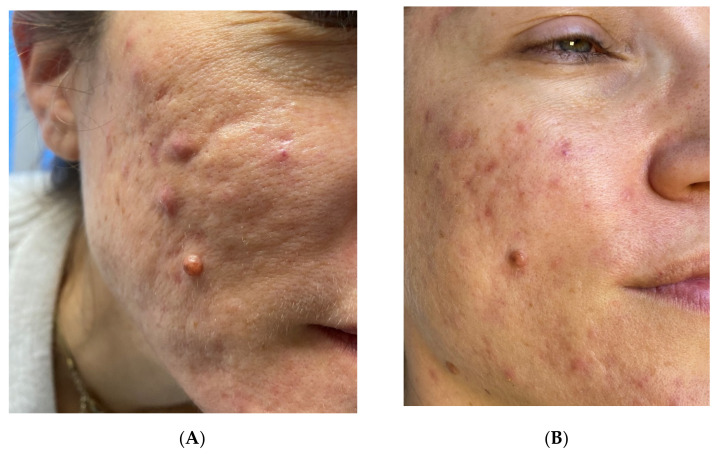
(**A**) Post acne atrophic scar. (**B**) After four sessions performed with LWP-RF device at 0,6watt energy and 3Hz frequency, with detached spots of sublimation performed on the edges around the scars, leaving columns of intact epidermis and sparing the central atrophic part, both to break down the edges and to stimulate the rise of the central skin depression.

**Figure 2 jcm-11-00140-f002:**
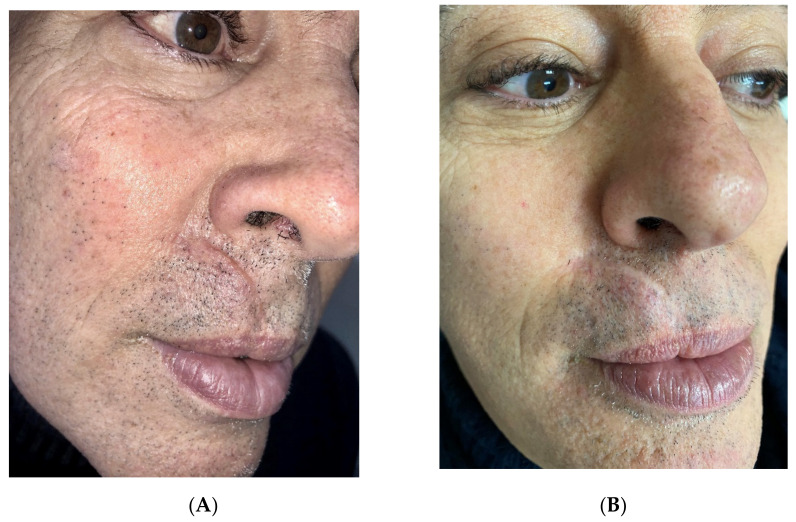
(**A**) Post traumatic hypertrophic scar. (**B**) After four sessions performed with LWP-RF device, with two sessions at 0,8 watt energy and 2Hz frequency, with no columns of sparing tissue, breaking down the thickening, and later with other two session at 0,6 watt energy and 3Hz frequency, with detached spots of sublimation to get the shrinkage, a reduction of thickness with a good settlement on the skin level, a reduction of erythema/pigmentation and a improvement of texture of the hypertrophic scar were observed.

**Figure 3 jcm-11-00140-f003:**
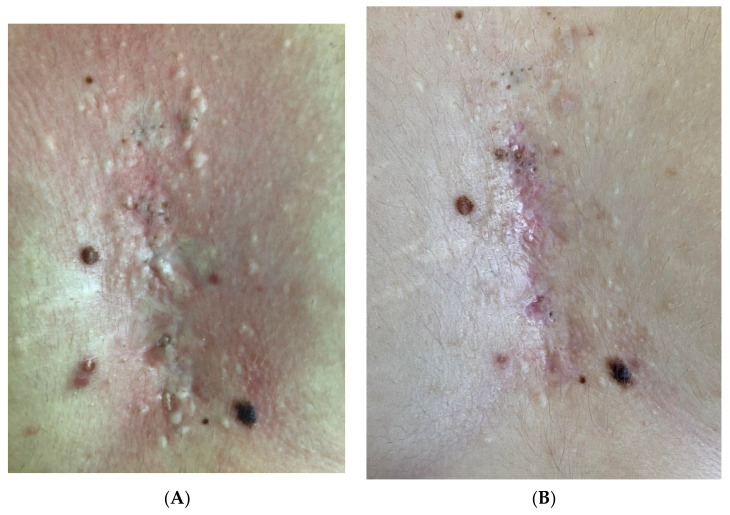
(**A**) Hidradenitis suppurativa atrophic/hypertrophic scar. (**B**) After four sessions performed with LWP-RF device at 0,6 watt energy and 3Hz frequency, breaking down the edges and stimulating the rise of the central skin depression, an improvement of atrophy was observed. Moreover a reduction of thickness and a improvement of texture of the hypertrophic parts of the scars treated with four session at 0,6 watt energy and 3Hz frequency, with detached spots of sublimation to get the shrinkage, were also observed.

## Data Availability

The study did not report any data.

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
