# Peer review of "Plasma Radiofrequency Ablation for Scar Treatment"

_jcm, 2021, doi:10.3390/jcm11010140_

Round 1
Reviewer 1 Report
Thank you for your great manuscript about "Plasma radiofrequency ablation for scar treatment"
It's very useful for readers.
If possible, please show the data of VAS change between pre and post treatment.
Author Response
Dear reviewer, thank you for the positive comments. As for the vas, its value at the starting point, considered as an unsatisfactory condition, was rated "0" by all patients. Cordially Adone BaroniReviewer 2 Report
The manuscript is well-written and the results highlight the key findings from this limited case series (the article should be labeled as such)... however, much improvement could be added to the Introduction and Discussion/Conclusions sections of the manuscript. For example, there should be more background information on the current standard of care for atrophic or hypertrophic scars with laser therapies. Furthermore, the Discussion lacks depth and breadth of discussing the general body of literature surrounding laser treatments (ablative and non-ablative), particularly the P-RF laser, which is the one used in this study. The authors must bolster this manuscript with sufficient discussion of their results in comparison to the results shown in prior studies already published in the literature, as well as a thorough description of this study's limitations and applications for future studies.
Author Response
Dear Reviewer,
You find attached to this message the manuscript corrected following your suggestions.
I hope it can be to your liking and satisfactory.
Cordially

Round 2
Reviewer 2 Report
Much improved on many of the points emphasized in initial critique.
As a minor addition, I would advise the authors add in the Limitations section of the Discussion the specific limitations of this present study (they listed a limitation of the laser therapy specifically by excluding patients with keloids... however, no mention of limitations of the present study were listed/discussed).
Otherwise, manuscript is much improved overall.
Author Response
Dear Reviewer,
following your suggestions, I have added the limitations of the study. Moreover I have also provided to the correction of English language and style by Papertrue online system.
Thank you very much
